# Cryo-EM structure of the benzodiazepine-sensitive α1β1γ2S tri-heteromeric GABA$_A$ receptor in complex with GABA

**Swastik Phulera[1†], Hongtao Zhu[1†], Jie Yu[1†], Derek P Claxton[1‡], Nate Yoder[1], Craig Yoshioka[1], Eric Gouaux[1,2]***

[1]Vollum Institute, Oregon Health and Science University, Portland, United States; [2]Howard Hughes Medical Institute, Oregon Health and Science University, Portland, United States

**\*For correspondence:**
gouauxe@ohsu.edu

[†]These authors contributed equally to this work

**Present address:** [‡]Department of Molecular Physiology and Biophysics, Vanderbilt University, Nashville, United States

**Abstract** Fast inhibitory neurotransmission in the mammalian nervous system is largely mediated by GABA$_A$ receptors, chloride-selective members of the superfamily of pentameric Cys-loop receptors. Native GABA$_A$ receptors are heteromeric assemblies sensitive to many important drugs, from sedatives to anesthetics and anticonvulsant agents, with mutant forms of GABA$_A$ receptors implicated in multiple neurological diseases. Despite the profound importance of heteromeric GABA$_A$ receptors in neuroscience and medicine, they have proven recalcitrant to structure determination. Here we present the structure of a tri-heteromeric α1β1γ2S$_{EM}$ GABA$_A$ receptor in complex with GABA, determined by single particle cryo-EM at 3.1–3.8 Å resolution, elucidating molecular principles of receptor assembly and agonist binding. Remarkable N-linked glycosylation on the α1 subunit occludes the extracellular vestibule of the ion channel and is poised to modulate receptor assembly and perhaps ion channel gating. Our work provides a pathway to structural studies of heteromeric GABA$_A$ receptors and a framework for rational design of novel therapeutic agents.
DOI: https://doi.org/10.7554/eLife.39383.001

## Introduction

GABA$_A$ receptors are chloride permeable, γ-amino butyric acid (GABA)-gated ion channels that are responsible for the majority of fast inhibitory neurotransmission in the mammalian nervous system (*Sigel and Steinmann, 2012*). Because of the fundamental role that GABA$_A$ receptors play in balancing excitatory signaling, GABA$_A$ receptors are central to the development and normal function of the central nervous system (*Wu and Sun, 2015*). In accord with their crucial role in brain function, mutations in GABA$_A$ receptor genes are directly linked to epilepsy syndromes (*Hirose, 2014*) and are associated with schizophrenia, autism, alcohol dependence, manic depression and eating disorder syndromes (*Rudolph and Möhler, 2014*). Moreover, GABA$_A$ receptors are the targets of a large number of important therapeutic drugs, from sedatives, sleep aids and anticonvulsant medications to anesthetic agents (*Braat and Kooy, 2015*). GABA$_A$ receptors are also the target of alcohol and are implicated in alcohol dependence (*Trudell et al., 2014*).

GABA$_A$ receptors belong to the pentameric ligand-gated ion channel (pLGIC) superfamily (*Thompson et al., 2010*). Other members of this family are nicotinic acetylcholine (nAChR), 5-HT$_{3A}$, glycine, and the invertebrate GluCl and Zn$^{2+}$-activated cation channels (*Thompson et al., 2010*). Members of the pLGIC superfamily are composed of five protein subunits and each subunit contains four transmembrane domains (M1–M4) along with extracellular N- and C- termini. GABA$_A$ receptors are typically found as heteromeric channels derived from a pool of 19 possible subunits: α1–6, β1–3, γ1–3, δ, ε, θ, π, and ρ1–3 (*Sigel and Steinmann, 2012*). The large number of subunits gives rise to

many possible pentameric assemblies; nevertheless, the most prevalent subunit combination in the vertebrate brain is the tri-heteromeric receptor composed of two α, two β and one γ subunit (*Chang et al., 1996*; *Farrar et al., 1999*; *Tretter et al., 1997*), with the arrangement of subunits being β-α-β-γ-α, in a clockwise order when viewed from the extracellular space (*Baumann et al., 2001*; *Baumann et al., 2002*; *Baur et al., 2006*). The molecular basis for selective subunit assembly of GABA_A receptors is not well understood.

Pioneering structural studies of the paradigmatic acetylcholine receptor (AChR) (*Unwin, 2005*), as well as crystallographic studies of homomeric pLGICs that include prokaryotic pLGICs (*Hilf and Dutzler, 2008*; *Sauguet et al., 2013*) and the eukaryotic GluCl (*Hibbs and Gouaux, 2011*), 5-HT_{3A} serotonin receptor (*Hassaine et al., 2014*), β3 GABA_A (*Miller and Aricescu, 2014*), α3 glycine receptor (GlyR) (*Huang et al., 2015*), along with the cryo-EM structures of the zebrafish α1 GlyR (*Du et al., 2015*) and the mouse 5-HT_{3A} receptor (*Basak et al., 2018*), have helped to shape our understanding of receptor architecture and mechanism. Recent structures of diheteromeric nAChRs also further our understanding of subunit arrangement and function in heteromeric Cys-loop receptors (*Walsh et al., 2018*).

These studies, together with a large number of biochemical and biophysical experiments, have defined the transmembrane ion channel pore, its lining by the M2 helices and likely mechanisms of ion selectivity (*Cymes and Grosman, 2016*; *Sine et al., 2010*). The extracellular N-terminal domain harbors the orthosteric agonist binding site, located at subunit interfaces, in addition to multiple binding sites for an array of small molecules and ions that act as allosteric modulators (*Lynagh and Pless, 2014*). While the mechanism by which orthosteric agonists and competitive antagonists activate or inhibit ion channel activity is well explored (*Gielen and Corringer, 2018*), the molecular mechanisms for the action of allosteric ligands, especially those that act on heteromeric GABA_A receptors, remain to be fully elucidated. Here we present methods for the expression and isolation of tri-heteromeric GABA_A receptors and the cryo-EM structure of the rat α1β1γ2S_EM GABA_A receptor in the presence of GABA. The structural analysis not only defines subunit organization, but it also uncovers the mode of GABA binding to the orthosteric agonist binding site and suggests a critical role of *N*-linked glycosylation of the α1 subunit in governing receptor assembly and, perhaps, ion channel activity.

## Results

### Receptor expression and structure elucidation

To enhance receptor expression we employed the M3/M4 loop deletion constructs of the α1 and β1 subunits analogous to the functional M3/M4 loop deletion constructs of GluCl (*Hibbs and Gouaux, 2011*) and GlyR (*Du et al., 2015*), together with a full-length construct of the γ2S (short splice variant) subunit (*Figure 1—figure supplement 1*), yielding the α1β1γ2S_EM construct (*Claxton and Gouaux, 2018*). Optimization of receptor expression constructs and conditions were monitored by fluorescence-detection, size-exclusion chromatography (FSEC) (*Kawate and Gouaux, 2006*). We included a 1D4 affinity tag (*MacKenzie et al., 1984*) on the γ2 subunit to selectively isolate the heteromeric complex from baculovirus-transduced mammalian cells (*Goehring et al., 2014*). For ensuing cryo-EM studies, we developed an α1 subunit-specific monoclonal antibody, 8E3, with the aim of using the Fab to identify the α1 subunit in the pseudo-symmetric receptor complex (*Figure 1—figure supplement 2*). The resulting purified α1β1γ2S_EM receptor, in the presence of the 8E3 Fab, binds muscimol, a high affinity agonist, and flunitrazepam, a benzodiazepine, with affinities similar to the full-length receptor (*Figure 1—figure supplement 3*) (*Hauser et al., 1997*; *Johnston, 2014*).

Moreover, the α1β1γ2S_EM receptor also exhibits ion channel gating properties that are similar to the wild-type receptor in the presence and in the absence of the 8E3 Fab (*Figure 1—figure supplement 3*) (*Li et al., 2013*). We note, however, that while the cryo-EM construct retains potentiation by diazepam, the extent of potentiation is reduced for the Fab complex.

Structure elucidation was carried out using the α1β1γ2S_EM receptor solubilized in β-dodecyl-maltoside (C12M) and cholesterol hemisuccinate (CHS) in the presence of 1.5 mM GABA. To enhance particle density on the cryo-EM grids in light of modest levels of receptor expression, we employed grids coated with graphene oxide. We proceeded to collect micrographs using a Titan Krios microscope and a Falcon three camera as described in the Materials and methods. Subsequent selection

of particles and calculation of 2D class averages yielded projections that were readily identified as a pentameric Cys-loop receptor bound by 2 Fabs (*Figure 1* and *Figure 1—figure supplement 4*). Three dimensional reconstruction, combined with judicious masking of either the Fab constant domains or, alternatively, the receptor transmembrane domain (TMD), allowed for reconstructions at ~3.8 and ~3.1 Å resolution, respectively, based on Fourier shell correlation (FSC) analysis (*Figure 1—figure supplement 5* and *Supplementary file 1*). We note that there is substantial preferred orientation in the particle distribution and, despite substantial efforts in exploring a wide spectrum of conditions, we were unable to obtain grids that yielded more well distributed particle orientations. Inspection of the resulting density maps were consistent with these resolution estimations, and in the case of the density in the extracellular domain (ECD), the quality of the density map is excellent, allowing for visualization of medium and large side chains, as well as glycosylation of Asn side chains (*Figure 1—figure supplement 6*). By contrast, the density for the TMD is not as well defined. While the M1, M2 and M3 helices of all subunits show strong density, with density for some residues with large side chains, the M4 helices of the two $\alpha$1 subunits and the $\gamma$2S subunit have weak density and thus we have not attempted to include them in the structure. To build a molecular structure into the cryo-EM density maps, we first generated homology models of the $\alpha$1, $\beta$1 and $\gamma$2S subunits using the human $\beta$3 GABA$_A$ receptor (PDB code: 4COF) (*Miller and Aricescu, 2014*) as a template and we then manually fit the models to the density and carried out iterative cycles of manual fitting and computational refinement, which together resulted in a structural model that fits well to the density and that has good stereochemistry (*Supplementary file 1*).

## Tri-heteromeric GABA$_A$ receptor subunit arrangement

The $\alpha$1$\beta$1$\gamma$2S$_{EM}$ receptor hews to the classic architecture of Cys-loop receptors, first established by cryo-EM studies of the nicotinic receptor (*Toyoshima and Unwin, 1988*; *Unwin, 1993*; *Unwin, 2005*), with a clockwise subunit arrangement of $\alpha$1-$\beta$1*-$\gamma$2S-$\alpha$1*-$\beta$1 when viewed from the extracellular side of the membrane. Here we label the $\alpha$1* and $\beta$1* subunits that are adjacent to the unique $\gamma$2S subunit with asterisks in order to distinguish them from their chemically equivalent yet spatially distinct $\alpha$1 and $\beta$1 partners. The arrangement of subunits in this heteromeric complex, as mapped out by the $\alpha$1-specific Fab fragments, is in agreement with previous biochemical studies (*Figure 1—figure supplement 2*) (*Baur et al., 2006*). The subunit identity is further verified by prominent *N*-linked glycosylation sites that are unique to each subunit. The epitope of the Fab resides entirely on the periphery of the $\alpha$1 ECDs and the Fab buries approximately 760 Å$^2$ of surface area in the interface with the receptor. Most of the receptor residues that interact with the Fab are located on the $\beta$8, $\beta$9 and $\beta$10 elements of secondary structure (*Figure 1—figure supplements 7* and *8*). While the Fab binding site is near the crucial C-loop, we note that it does not overlap with it and thus the Fab is unlikely to directly influence agonist binding, in agreement with the agonist binding experiments (*Figure 1—figure supplement 3*).

## Subunit interfaces

Subunit-subunit interactions within the extracellular domain play a major role in pLGIC function (*Jones and Henderson, 2007*) and in GABA$_A$ receptor assembly. Here, we estimate that within the ECD each subunit buries as much as 1150–1600 Å$^2$ of solvent accessible surface area within each subunit interface (*Supplementary file 2*). There are five unique subunit interfaces in the tri-heteromeric receptor and, while we observed a similar arrangement of the ECDs at each interface, we also found subtle differences due to variations in amino acid sequences and in local protein structure. There are solvent accessible fenestrations at subunit interfaces in the ECD that 'connect' the extracellular solution with the extracellular vestibule, possibly providing an alternative pathway for ions to access the ion channel (*Figure 2A*). When comparing the various subunit interfaces one can readily identify specific interactions of amino acids that are common to all interfaces (*Figure 2*). As examples, at the $\alpha$(+)/$\beta$(-) interface (*Figure 2B*), there is a conserved interaction between Tyr209 at the beginning of $\beta$10 from the $\alpha$ subunit at the (+) face to Arg117 at $\beta$5 in (–) face, as well as a hydrogen bond between Tyr206 from the $\alpha$ subunit and Gln64 from the $\beta$ subunit. At the $\beta$(+)/$\alpha$*(-) interface (*Figure 2C*), a similar interaction is observed between Tyr205 at the beginning of $\beta$10 from the $\beta$ subunit of the (+) subunit and Arg117 at $\beta$5 in the (–) subunit. Furthermore, a hydrogen bond between Thr202 from the $\beta$ subunit and Arg66 from the $\beta$ subunit is observed even though, in

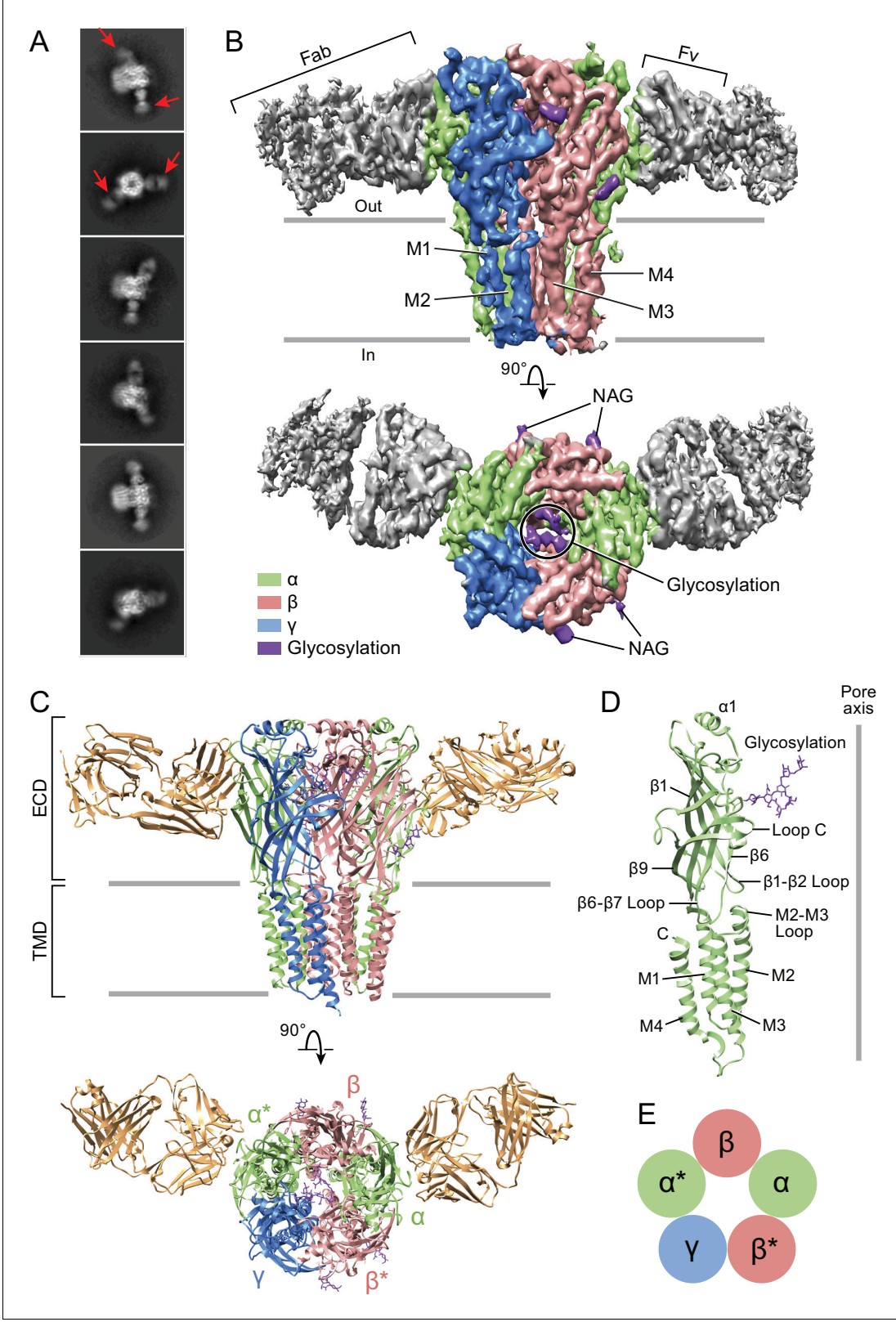

**Figure 1.** Architecture of the α1β1γ2S$_{EM}$ GABAA receptor. (a) 2D class averages. Red arrows indicate 8E3 Fab bound to α subunits. (b) The cryo-EM map of the entire receptor viewed parallel to membrane plane. The α, β and γ subunits are colored by lime, salmon and marine, respectively. (c) Cartoon representation of the receptor viewed parallel to the membrane plane. The extracellular domain (ECD) and transmembrane domain (TMD) are indicated.

*Figure 1 continued on next page*

*Figure 1 continued*

(d) Cartoon representation of an α subunit. (e) Schematic representation of subunit arrangement, viewed from the extracellular side of the membrane.

DOI: https://doi.org/10.7554/eLife.39383.002

The following figure supplements are available for figure 1:

**Figure supplement 1.** Schematics for the constructs used in this study.

DOI: https://doi.org/10.7554/eLife.39383.003

**Figure supplement 2.** Representative FSEC traces showing the subunit specificity of the 8E3 antibody.

DOI: https://doi.org/10.7554/eLife.39383.004

**Figure supplement 3.** Analysis of α1β1γ2S$_{EM}$ receptor function.

DOI: https://doi.org/10.7554/eLife.39383.005

**Figure supplement 4.** The processing workflow of cryo-EM data analysis of tri-heteromeric GABA$_A$ receptor data set.

DOI: https://doi.org/10.7554/eLife.39383.006

**Figure supplement 5.** Cryo-EM analysis of the tri-heteromeric data set.

DOI: https://doi.org/10.7554/eLife.39383.007

**Figure supplement 6.** Representative densities for the 'ECD' and 'whole' cryo-EM maps.

DOI: https://doi.org/10.7554/eLife.39383.008

**Figure supplement 7.** Sequence alignment of the α1, β1, γ2S sequence with secondary structure assignment marked.

DOI: https://doi.org/10.7554/eLife.39383.009

**Figure supplement 8.** 8E3 fab binding to the α1β1γ2S$_{EM}$ receptor.

DOI: https://doi.org/10.7554/eLife.39383.010

comparison to the α(+)/β(-) interface, the Gln is replaced by Arg. At the α*(+)/γ(-) interface (*Figure 2D*), the distance between Tyr209 from the α and Arg132 from the γ subunit is similar to

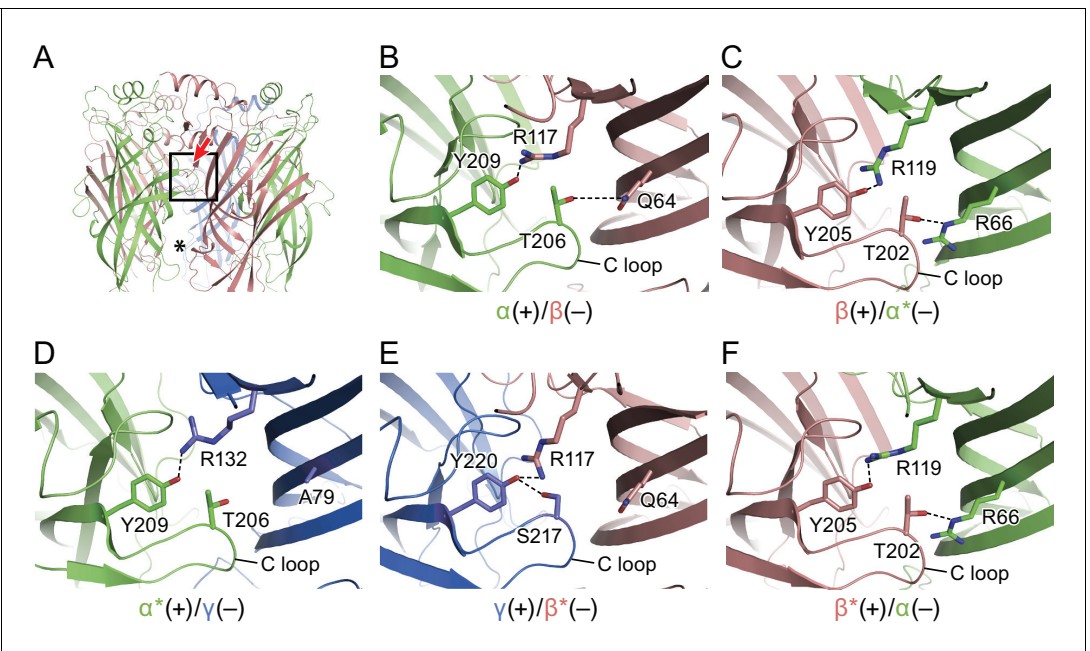

**Figure 2.** Intersubunit interactions. (a) The interface region between the α and β subunits, with the boxed area enlarged in panels b-f and the viewing angle indicated by an arrow, the region with fenestration between the subunits is marked with an asterisk. The α and α* subunits are colored in salmon, β and β* are colored in lime, and γ is colored in marine and shown as cartoon. Side chains for amino acids for which specific interactions were observed are shown with interactions depicted in dashed lines. In (b) is the α(+)/β(-) interface, (c) the β(+)/α*(-) the interface, (d) the α*(+)/γ(-) interface, (e) the γ(+)/β*(-) interface and (f) the β*(-)/α(-) interface.

DOI: https://doi.org/10.7554/eLife.39383.011

that observed at the α(+)/β(-) interface. The hydrogen bond observed in α(+)/β(-), β(+)/α*(-), β*(+)/α (-) (*Figure 2B,C and F*) between a Thr from the (+) face and Arg/Gln at the (-) face is lost because Ala79 occupies the position corresponding to the Arg/Gln residues. At the γ(+)/β*(-) interface, Tyr220 from the γ subunit is within hydrogen bonding distance to Arg117 from the β subunit. Interestingly, with the other set of interactions seen in the α(+)/β(-), β(+)/α*(-) and β*(+)/α(-) interfaces (*Figure 2B,C and F*) between a Thr from the (+) face and a Arg/Gln at the (–) face, the Thr is replaced by a Ser and forms a hydrogen bond with the Tyr220 from the γ subunit rather than interacting with the Gln64 from the β subunit. The β*(-)/α(-) interface closely resembles the spatially distinct β(+)/α*(-) interface (*Figure 2F,C*), with both of the interactions preserved.

To investigate the overall conformation of the extracellular domain, we compared the ECDs of our current structure and their relative positioning to the existing structures of homomeric pLGICs. To do this, we superposed the α subunit of our structure onto one of the subunits of the homomeric 5-HT$_{3A}$ receptor in the apo (*Basak et al., 2018*) or the desensitized state (*Hassaine et al., 2014*), the human GABA$_A$ receptor bound to benzamidine (*Miller and Aricescu, 2014*) and homomeric GlyR bound to strychnine or to glycine/ivermectin (*Du et al., 2015*) (*Figure 3—figure supplement 1*).

We observe that the ECDs in all the homomeric structures are located at similar distances as in our current structure. Nevertheless, if we compare the pentagon formed by joining a line through the center of mass of these individual ECDs, we observe that these are rotated in comparison to our current structure and that our current structure most resembles the conformation of glycine/ivermectin-bound GlyR. Thus, we propose that the conformation of the ECD represents an agonist/allosteric modulator bound, activated-like state.

## Neurotransmitter binding sites

To illuminate the molecular basis for GABA binding, we determined the structure of the α1β1γ2S$_{EM}$ receptor in the presence of saturating GABA (*Sigel and Steinmann, 2012*). Neurotransmitter binding sites in Cys-loop receptors are located at the interface of two adjacent subunits and are composed of the three loops from the principle (+) face and β-strands from the complementary (-) face (*Nys et al., 2013*). There are three substantive densities within the neurotransmitter binding sites, at the interface between the β*(+)/α(-), the α(+)/β(-) and the β(+)/α*(-) subunits, an observation that diverges from previous studies suggesting that there are only two canonical GABA binding sites located at the β*(+)/α(-) and β(+)/α*(-) interfaces (*Figure 3A* and *Figure 3—figure supplement 2*) (*Chua and Chebib, 2017*). Nevertheless, other studies have pointed out that GABA may bind to interfacial binding sites in addition to the two canonical sites (*Chua and Chebib, 2017*). The oval-like densities in the two canonical sites are well fit by the chemical structure of GABA, although the density feature at the β(+)/α*(-) interface is weaker than that at the β*(+)/α(-) site (*Figure 3—figure supplement 2*). Interestingly, the third feature at the α(+)/β(-) interface, with a sausage-like shape, has the strongest density (*Figure 3—figure supplement 2*). Because GABA is the only small molecule present in the sample buffer that has a size and shape similar to the density feature, we speculate that the density belongs to a GABA molecule. Nevertheless, it is possible that the density at the α(+)/β(-) interface belongs to an unidentified small molecule that co-purified with the receptor. Alternatively, the density feature could be due to several ordered water molecules. Given the continuous nature of the density, however, we favor the notion that the density is attributable to non-water, small molecule. Further studies will be required to experimentally determine GABA binding stoichiometry, and higher resolution cryo-EM studies will be needed to more thoroughly define the density features.

In the canonical binding sites, GABA is wedged between the β*(+) and α(-) subunits with extensive interactions with residues from loop C, loop B, loop A, β2 strand and β6 strand. The amino group of GABA likely forms hydrogen bonds with the backbone carbonyl oxygen of Tyr157 (loop B), Glu155 (loop B) and Tyr 97 (loop A) and a cation-π interaction with Tyr205, while the carboxylate group forms possible hydrogen bonds with Thr129 (β6 strand) and Thr202 (loop C) and a salt bridge with Arg66 (β2 strand). In addition, sandwiching of the amino group of GABA between Tyr205 (loop C) and Tyr157 further increases the number of interactions between agonist and receptor (*Figure 3B*). Notably, Tyr97, Glu155 and Arg66 are unique in the β subunit compared to the corresponding residues in the α and γ subunits and are crucial for substrate binding, as reported in previous studies demonstrating that Tyr97 and Arg66 play an important role in the binding pocket

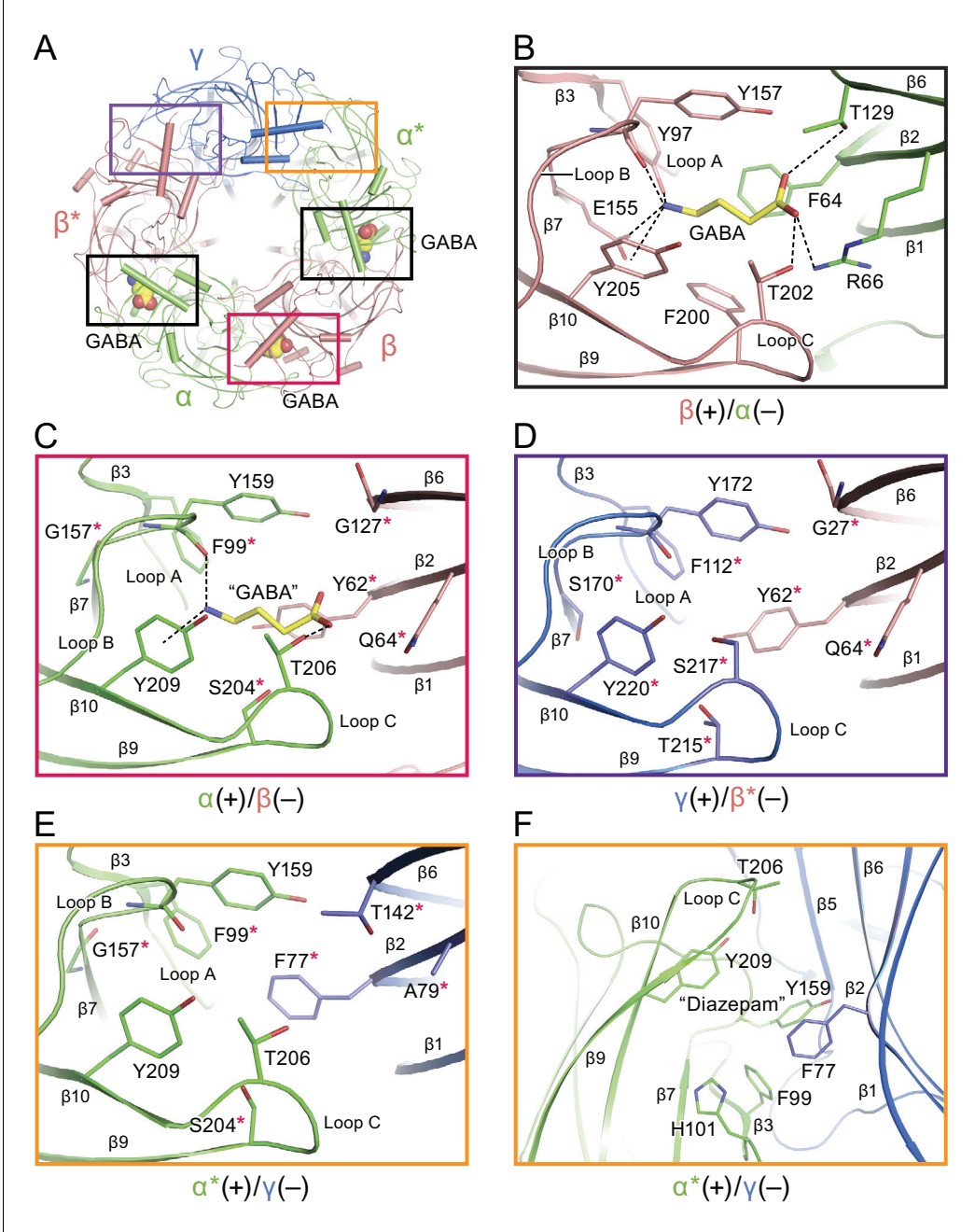

**Figure 3.** Neurotransmitter binding sites. (a) Top down view of the receptor looking from the extracellular side. The α and α* subunits are colored in salmon, β and β* are colored in lime, and γ is colored in marine. GABA molecules are shown in sphere representation. (b) View of the binding site between the β*(+)/α(-) subunits viewed parallel to the membrane. Dashed lines indicate hydrogen bonds, cation-π interactions and salt bridges. The β*(+) and α(-) subunits are colored in salmon and lime, respectively. The residues in the β*(+) and α(-) subunits and GABA are depicted in salmon, lime and yellow sticks, respectively. (c) View of the binding site between the α(+)/β(-) subunits viewed parallel to the membrane. Subunits and residues are depicted in the same color code as in (b). The residues differing from the corresponding residues in the β*(+)/α(-) binding site are indicated with red stars. (d) View of the binding site between the γ (+)/β*(-) subunits looking parallel to the membrane. Residues in the γ(+) and β*(-) binding site are shown in marine and salmon sticks, respectively. The residues differing from the corresponding residues in the β*(+)/α(-) binding site are indicated with red stars. (e) View of the binding site between the α*(+)/γ(-) binding site viewed parallel to the membrane. Residues in the α*(+) and γ(-) binding site are shown in lime and marine sticks, respectively. The residues differing from the corresponding residues in β*(+)/α(-) are indicated with red stars. (f) Similar view of the diazepam binding site as in panel (e).

*Figure 3 continued on next page*

*Figure 3 continued*

DOI: https://doi.org/10.7554/eLife.39383.012

The following figure supplements are available for figure 3:

**Figure supplement 1.** Superposition of the tri-heteromeric GABA$_A$ receptor ECD structure with the 5-HT$_{3A}$ receptor structure (PDB code: 4PIR and 6BE1), the homo GABA$_A$ β3 structure (PDB code: 4COF), the strychnine bound glycine receptor structure (PDB code: 3JAD), and the ivermectin-glycine bound glycine receptor structure (PDB code: 3JAF).

DOI: https://doi.org/10.7554/eLife.39383.013

**Figure supplement 2.** Densities surrounding the GABA binding pocket.

DOI: https://doi.org/10.7554/eLife.39383.014

**Figure supplement 3.** Superposition of the ECD between α, β* and γ subunits.

DOI: https://doi.org/10.7554/eLife.39383.015

**Figure supplement 4.** Superposition of the ECD of β*/ α subunits with the structure for human β3 GABA$_A$ (PDB code: 4COF), GlyR-open (PDB code: 3JAE) and GlyR-closed (PDB code: 3JAD) to illustrate the configuration of loop C in each structure.

DOI: https://doi.org/10.7554/eLife.39383.016

(*Newell et al., 2004*; *Sander et al., 2011*; *Smith and Olsen, 1995*). Experiments employing non native amino acids suggest that Tyr 97 forms a cation-π interaction with GABA instead of Tyr 205 (*Padgett et al., 2007*). We note, however, that the aromatic ring of Tyr 97 is sandwiched between Arg 131 of the adjacent α subunit and Glu 155 of β and thus, in this structure, it is not well positioned to form a cation-π interaction with GABA. Nevertheless, we cannot exclude the possibility that there are changes in the conformation of the binding pocket that allow for cation-π interactions between GABA and Tyr 97. Mutation of Tyr157, a highly conserved residue in the α, β and γ subunits, significantly reduces the binding affinity for agonist and antagonist (*Lummis, 2009*). Similar neurotransmitter binding interactions have previously been reported in the other Cys-loop members, including GluCl and GlyR (*Du et al., 2015*; *Hibbs and Gouaux, 2011*). Superposing this canonical binding site with the putative third 'GABA' binding site at the interface of the α(+)/β(-) site shows that there are fewer interactions between 'GABA' and the surrounding residues, such as the cation-π interactions with Tyr209 and contacts with Thr206 and the carbonyl oxygen of Tyr159 (*Figure 3C*). We thus emphasize that, while the density feature is unambiguous, the identification of this site as a bona fide GABA binding site will require additional investigation.

We also superposed the canonical GABA binding site at the β*(+)/α(-) interface with the homologous sites at the γ(+)/β(-) and α*(+)/γ(-) interfaces, finding differences among the residues that play important roles in binding GABA, differences that may lead to weak or no GABA binding (*Figure 3D and E*). In inspecting these sites that include the γ2S subunit, we have attempted to better understand the binding site of benzodiazepines, such as diazepam, which function as allosteric potentiators of the γ-subunit containing GABA$_A$ receptors (*Figure 1—figure supplement 3*) (*Li et al., 2013*). Previous studies demonstrated that diazepam binds to the α*(+)/γ(-) interface with high affinity (*Li et al., 2013*). In agreement with previous studies, we speculate that α*His101 could provide aromatic or hydrophobic interactions with the pendant phenyl ring of diazepam. Additionally, α*Tyr209, α*Phe99, α*Tyr159 and γPhe77 likely also contribute to hydrophobic interactions with diazepam (*Figure 3F*). All these key structural residues surrounding the diazepam site have been identified in homology models and functional experiments (*Richter et al., 2012*; *Teissére and Czajkowski, 2001*; *Wongsamitkul et al., 2017*). Indeed, mutation of α*His101, α*Tyr209 and α*Tyr159 markedly impairs the modulation by diazepam (*Amin et al., 1997*), while mutation of γPhe77 results in decreased binding affinity.

Agonist binding has been proposed to induce loop C closure in the open, ion conducting state and antagonist binding to stabilize an 'open' configuration of loop C in the closed, non-conducting state of the ion channel (*Du et al., 2015*; *Mukhtasimova et al., 2005*; *Purohit and Auerbach, 2013*). To probe the relationship between agonist binding and the position of loop C, we superposed the β* subunit with GABA bound onto the α* and γ subunits and found that loop C in the β* subunit is in an 'open' conformation relative to that in the α* and γ subunits (*Figure 3—figure supplement 3*). Additionally, superposition of the ECD of the β*/α subunit with the human β3 GABA$_A$, GlyR-open and GlyR-closed structures suggests that the position of loop C in the β* subunit closely

approximates that in the GlyR-closed non-conducting state, while loop C in the human β3 GABA$_A$ is in a more 'open' configuration (*Figure 3—figure supplement 4*). However, this interpretation is subject to caveats because it is derived from the comparison of different subunits and different receptors. It would be more persuasive to define the relationship between loop C and the functional state of the receptor using the same receptor in different ligand bound/functional states, as well as in the absence of bound Fab.

## Glycosylation within the extracellular vestibule

There is a well-defined glycosylation site at Asn110 of the α subunit, within the extracellular vestibule of the receptor, at a site of post-translational modification that is distinct from the other GABA$_A$ receptor subunits (*Figure 4A*) and not observed before in any Cys-loop receptor (*Figure 1—figure supplement 8*, *Figure 4—figure supplement 1*). This site is predicted to be glycosylated based on sequence analysis (*Blom et al., 2004*; *Julenius et al., 2005*; *Miller and Aricescu, 2014*), and the quality of the cryo-EM map allows us to confidently locate sugar residues involved in glycosylation (*Figure 4B and C*). For the α subunit at Asn110 we have built a carbohydrate chain with seven sugar residues and, for the α* subunit at Asn 110, we have built a carbohydrate chain with four sugar groups (*Figure 4C and D*). These two carbohydrate chains are remarkably well ordered, a feature that is likely due to the fact that the chains are contained within the extracellular vestibule and are in direct contact with numerous protein side chains (*Figures 4A* and *5*). Additional glycosylation sites found in the β subunit have been previously observed in the β3 crystal structure (*Miller and Aricescu, 2014*) and are located on the external surface of the receptor, pointing away from the receptor (*Figure 4B*).

The glycosylation at the α subunit is well supported by the density to the extent that even chain branches can be easily defined (*Figure 4C and D*). These glycosylation sites are located in the ECD at the center of the pore (*Figures 4B* and *5*) and they occupy a substantial portion of the ECD cavity. The volume of glycosylation in the vestibule is approximately 700 Å$^3$ at the contour level of 6 σ, which occupies nearly 18% of the total volume of the ECD vestibule. Because it is likely that each carbohydrate chain has sugar residues that are not resolved in the density map, we predict that the volume of the ECD vestibule occupied by these carbohydrate groups is actually larger than our estimation. There are extensive and specific interactions observed between the sugar groups and the γ2S subunit involving residues Asn101, Lys112 and Trp123 (*Figures 4A* and *5*).

We propose that the glycosylation at Asn110 of the α subunits is important for subunit assembly, effectively blocking the formation of pentamers with more than two α subunits, due to the fact that more than two carbohydrate chains would engage in sterically prohibitive van der Waals clashes. Moreover, we speculate that the contacts between the carbohydrate at position Asn110 of the α subunit and, for example, the Trp123 of the γ2S subunit, favor the inclusion of the γsubunit as the last subunit in the formation of the heteropentamer. This is consistent with prior studies which showed that glycosylation of the α subunit is essential for proper receptor assembly in mammalian cells (*Buller et al., 1994*), as well as the important role of tryptophan residues in interacting with carbohydrates in general (*Maenaka et al., 1994*; *Stewart et al., 2008*). We further suggest that these carbohydrate chains within the extracellular vestibule may also modulate receptor gating and, perhaps, ion channel block by large ion channel blockers. Indeed, it is intriguing that a post-translational modification such as N-linked glycosylation occupies such an important and 'internal' site in a neurotransmitter-gated ion channel.

## Conformation and asymmetry of the TMD

In the GABA$_A$ structure, the intrinsic flexibility of the TMD or the presence of the detergent micelle prevents us from accurately placing residues in each TM helix, especially in the M4 helices. Nevertheless, the density of the M1, M2 and M3 helices is well defined, allowing us to reliably position the helices. We note that in this receptor composed of α, β and γ subunits there is a breakdown in 5-fold symmetry observed in the homomeric or di-heteromeric Cys-loop receptors (*Figure 6*) (*Miller and Aricescu, 2014*; *Nys et al., 2013*). While the distances between the center of mass (COM) of two adjacent subunits are similar, where the COMs were calculated using the M1, M2 and M3 helices of each subunit, varying from 17.6 Å to 19 Å, the angles in the pentagon range from 94° to 124° (*Figure 6A*). Looking at the top-down views of five central M2 helices, the two α M2 helices

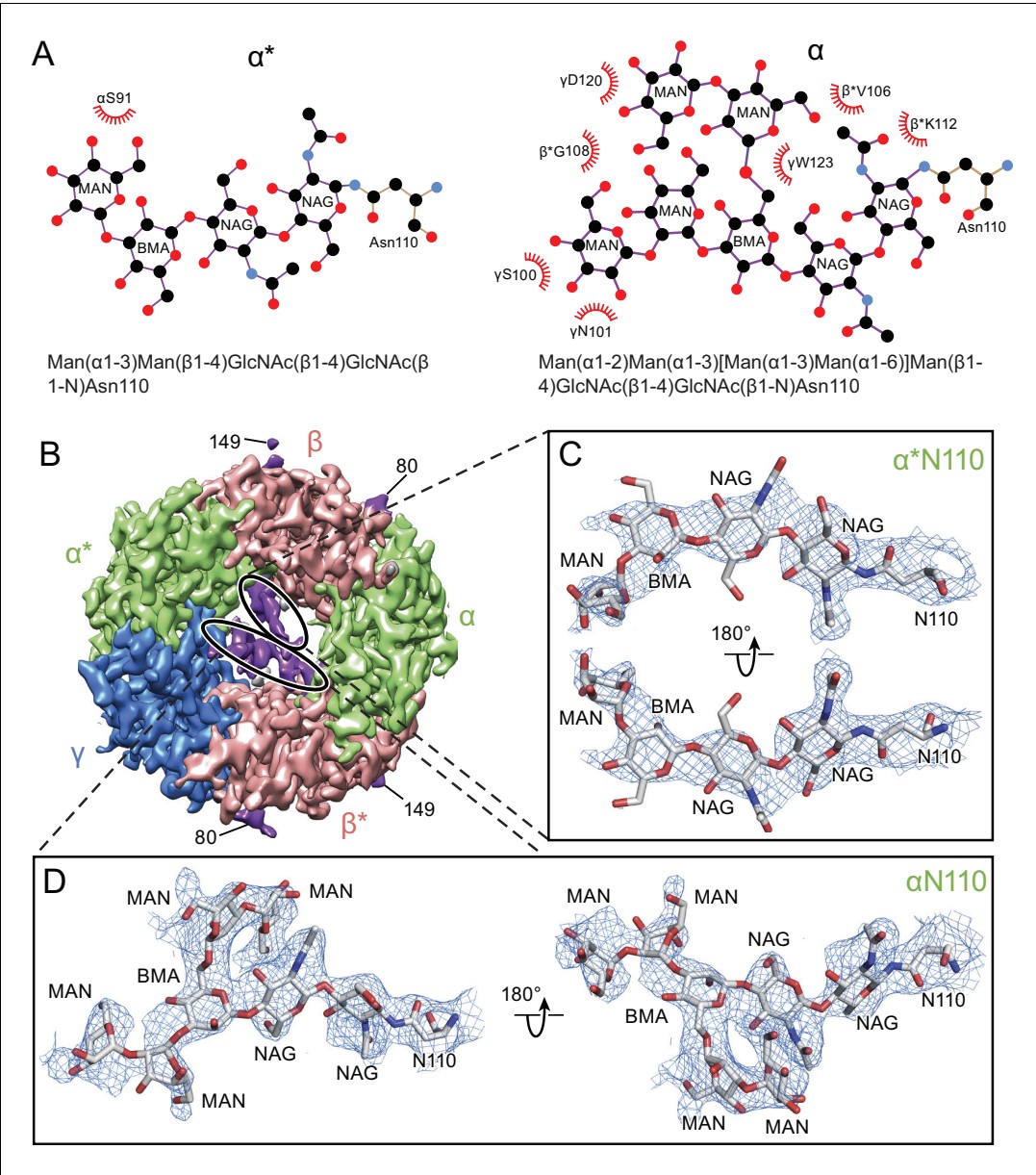

**Figure 4.** Glycosylation sites in the extracellular vestibule. (a) Schematic of the sugar chain chemical structure for the Asn110 modification. The amino acids interacting with the sugar chains are also shown. The names of the carbohydrates are given at the bottom of the panel (*Gamian, 1992*). (b) Top down view of ECD map. The α, β and γ subunits are colored by light green, salmon and blue, respectively. The glycosylation densities are colored by purple. The related Asn residue numbers are labeled. (c) Two views of the density of the glycosylation from the α* subunit, isolated and fitted with four sugar molecules. Asn110 and the name of sugars are labeled. (d) Similar panel to (c), showing the density of the glycosylation on the α subunit fit with a carbohydrate chain containing seven sugar residues.

DOI: https://doi.org/10.7554/eLife.39383.017

The following figure supplement is available for figure 4:

**Figure supplement 1.** Sequence alignment of the human Cys-loop receptors.
DOI: https://doi.org/10.7554/eLife.39383.018

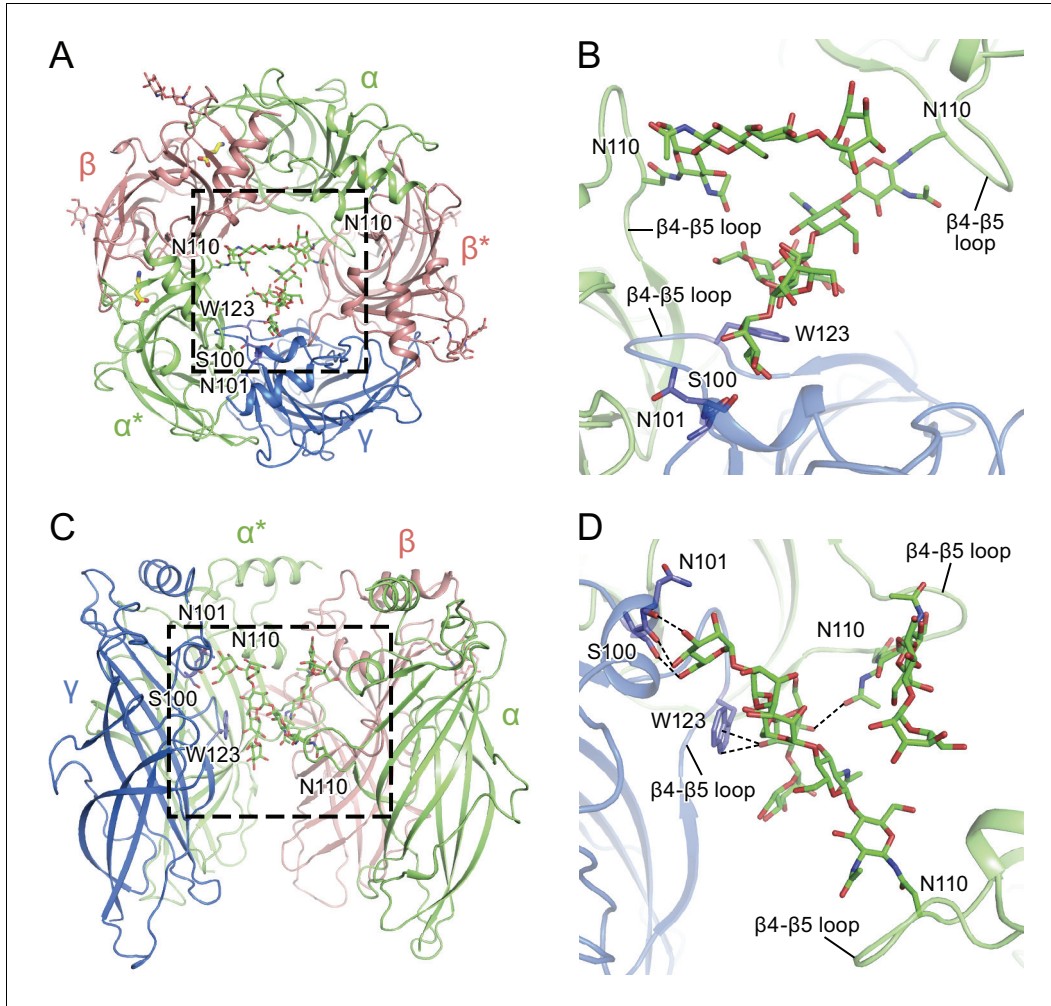

**Figure 5.** α-Subunit glycosylation and interaction with key residues. (**a**) Top-down view of the glycosylation pocket parallel to the membrane. The α and α* subunits are colored in salmon, the β and β* subunits are in lime and the γ subunit is in marine. Glycosylation from the α/α* and β/β* subunits are shown in green and salmon sticks. Interacting residues from the γ subunit are depicted in stick representation (marine). (**b**) Enlarged view of the glycosylation pocket indicated by the dash-outlined frame in (**a**). The β and β* subunits are removed for clarity. (**c**) Side view of the glycosylation pocket perpendicular to the membrane. The β* subunit was removed for clarity. (**d**) Enlarged view of the glycosylation pocket from the dash-outlined frame in (**c**). The β and β* subunits were removed for clarity.

DOI: https://doi.org/10.7554/eLife.39383.019

are located away from the other three TM helices. Superposition of the two β subunits with the human β3 GABA$_A$ receptor shows that there are rotations of each TM helix in the α1β1γ2S$_{EM}$ receptor structure relative to corresponding TM helices in the human β3 GABA$_A$ receptor, further demonstrating the asymmetric structure of the TMD (*Figure 6B*).

The asymmetry in the TMD complicates defining whether the structure represents an open or closed state, as the M2 helices lining the pore in the β subunit display a similar orientation as in the desensitized state of the human β3 GABA$_A$ receptor, yet the α subunits are farther from the pseudo 5-fold axis, enlarging the pore (*Figure 6B*). Because of the asymmetric structure of the TMD and the fact that the structure was solved using a receptor embedded in a detergent micelle, the TMD structure may represent a non-native conformation. Future structural studies using different detergents and lipids, or using lipid-filled nanodiscs, will be important in elucidating structures in specific functional states.

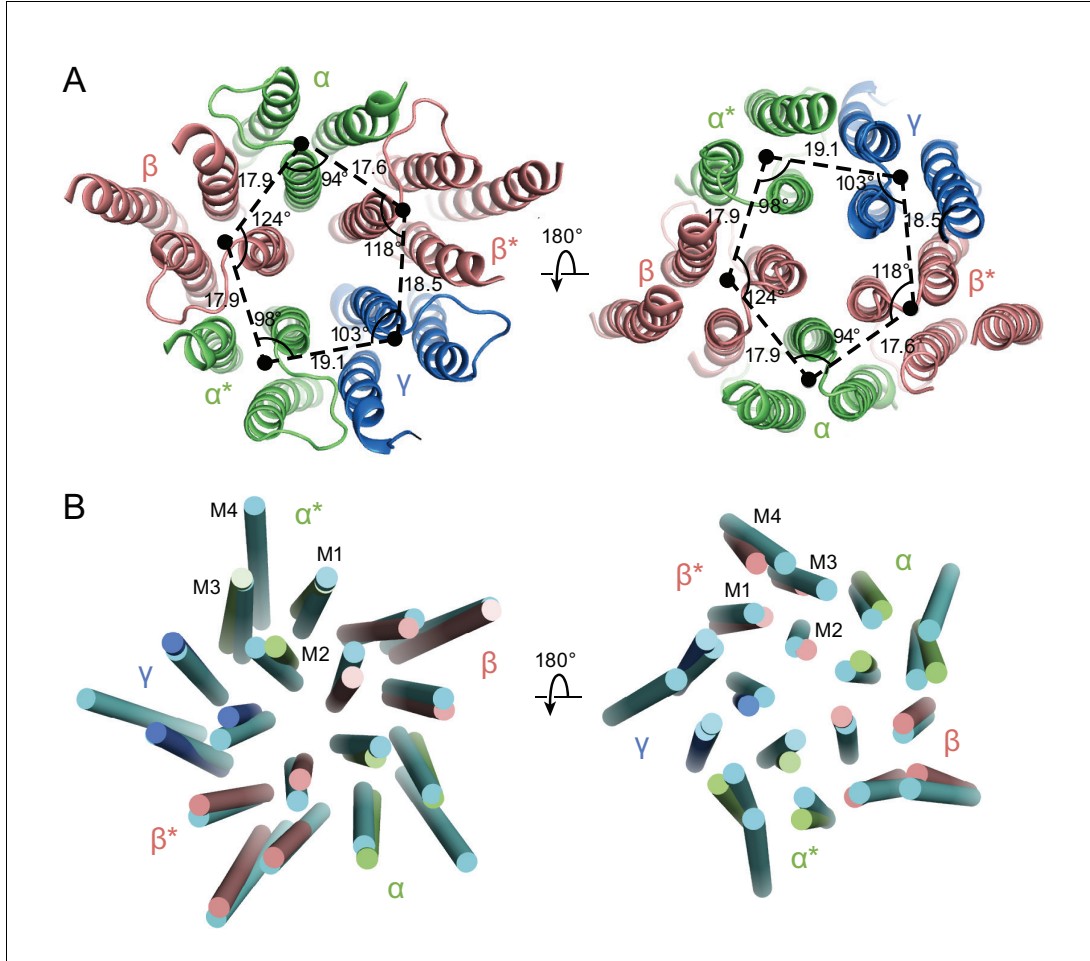

**Figure 6.** Asymmetry in the TMD. (**a**) View of the TMD from the extracellular (left panel) side or from the intracellular side (right panel) of the membrane. The α and α* subunits are colored in salmon, β and β* are colored in lime, and γ is colored in marine. Center of mass of M1, M2 and M3 helices for each subunit, shown as a solid black circle, was generated by Pymol. Distances are in Å. (**b**) Superposition of the TMD in the GABA$_A$ receptor with human β3 (PDB code: 4COF). The TMD of the human β3 is colored in cyan.

DOI: https://doi.org/10.7554/eLife.39383.020

## Discussion

Here we directly visualize the subunit stoichiometry and subunit arrangement for a tri-heteromeric GABA$_A$ receptor. While previous studies have suggested that GABA$_A$ receptor assembly occurs via defined pathways that limit the receptor diversity (*Sarto-Jackson and Sieghart, 2008*), precise molecular mechanisms regarding assembly are unknown. If we consider how heteromeric GABA$_A$ receptors form from α, β and γ subunits, we speculate that the larger contact areas in the ECD region makes formation of the β(+)/α(-) interface favorable, thus leading to α/β assembly intermediates, which then transition to (α/β)$_2$ intermediates. The crucial glycosylation site on the α subunit at Asn110 disfavors addition of a third α subunit, due to steric clashes, and that either a β or a γ subunit will be incorporated to complete the heteropentamer (*Figure 7*). Thus, the glycosylation at Asn110 of the α subunits not only plays an important role in ion channel assembly, but it may also be important to ion channel function.

Investigation of the GABA binding site identified subunit-specific interactions. Interestingly, we also find a strong density at a non-canonical site at the α(+)/β(-) interface. We postulate that this density is GABA, although it is also possible that it represents a ligand that co-purified with the receptor and could be a binding site for agonists or other small molecules that are involved in regulation of the receptor function. Our results lay the foundation for future studies directed towards

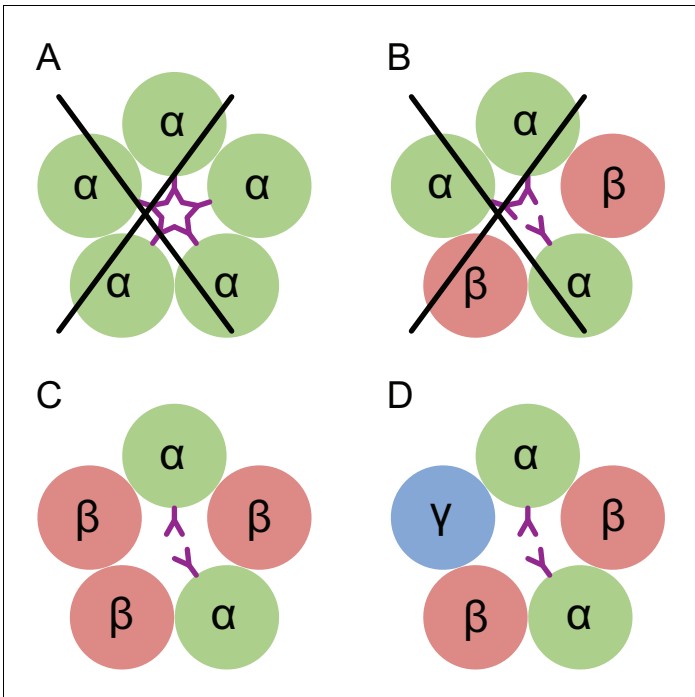

**Figure 7.** Conceptual schematic of tri-heteromeric receptor assembly. Steric clashes prevent the formation of a pentameric receptor with more than 2 α subunits, while other combinations are allowed. Individual subunits are marked and shown as circles. The glycosylation of the α subunit at position Asn110 is shown as a purple 'Y'.
DOI: https://doi.org/10.7554/eLife.39383.021

developing novel therapeutic agents that modulate GABA$_A$ receptors, provide methods to express and elucidate structures of GABA$_A$ receptors and illuminate the role of post translational modifications on GABA$_A$ receptor assembly, structure and function.

# Materials and methods

**Key resources table**

| Reagent type (species) or resource | Designation | Source or reference | Identifiers | Additional information |
|---|---|---|---|---|
| Gene (*Rattus norvegicus*) | GBRB1_RAT | Synthetic | UniProtKB - P15431 | |
| Gene (*Rattus norvegicus*) | GBRA1_RAT | Gift from Dr. David S. Weiss | UniProtKB - P62813, Gene ID 29705 | |
| Gene (*Rattus norvegicus*) | GBRG2_RAT | Gift from Dr. David S. Weiss | UniProtKB - P18508, Gene ID 29709 | |
| Cell line (Homo Sapiens) | TSA201 | ATCC | ATCC Cat# CRL-11268, RRID:CVCL_1926 | |
| Antibody | 8E3 | OHSU VGTI, Monoclonal Antibody Core | | Isotype: IgG2b, kappa |
| Recombinant DNA reagent | pEG BacMam | doi: 10.1038/nprot.2014.173 | | |
| Chemical compound, drug | GABA | Sigma | CAT NO. A5835 | |
| Software, algorithm | Relion-2.1 | doi: 10.1016/j.jsb.2012.09.006 | RRID:SCR_016274 | https://www2.mrc-lmb.cam.ac.uk /relion/index.php?title=Main_Page |
| Software, algorithm | MotionCor2 | doi:10.1038/nmeth.4193 | | http://msg.ucsf.edu/em/software/ motioncor2.html |

*Continued on next page*

*Continued*

| Reagent type (species) or resource | Designation | Source or reference | Identifiers | Additional information |
|---|---|---|---|---|
| Software, algorithm | Pymol | PyMOL Molecular Graphics System, Schrödinger, LLC | RRID:SCR_000305 | http://www.pymol.org/ |
| Software, algorithm | UCSF Chimera | doi:10.1002/jcc.20084 | RRID:SCR_004097 | http://plato.cgl.ucsf.edu/chimera/ |
| Software, algorithm | Bsoft | doi:10.1006/jsbi.2001.4339 | | https://lsbr.niams.nih.gov/bsoft/ |
| Software, algorithm | Localscale | doi:10.7554/eLife.27131 | | https://git.embl.de/jakobi/LocScale |
| Software, algorithm | Phenix | doi:10.1107/S2059798318006551 | RRID:SCR_014224 | https://www.phenix-online.org/ |
| Software, algorithm | Molprobity | doi:10.1107/S0907444909042073 | RRID:SCR_014226 | http://molprobity.biochem.duke.edu |
| Software, algorithm | cryoSparc | doi:10.1038/nmeth.4169 | | https://cryosparc.com/ |
| Software, algorithm | Gctf | doi:10.1016/j.jsb.2015.11.003 | | https://www.mrc-lmb.cam.ac.uk/kzhang/Gctf/ |
| Software, algorithm | DoG picker | doi: 10.1016/j.jsb.2009.01.004 | | https://omictools.com/dog-picker-tool |

## Construct description

The full-length rat $GABA_A$ receptor subunit isoforms α1 (Gene ID 29705) and γ2S (Gene ID 29709) with the native signal sequence were a gift from Dr. David S. Weiss. The α1-LT construct was generated by replacing a portion of the M3/M4 cytoplasmic loop from 313 to 381 by a Gly-Thr linker and adding a thrombin site and a $His_8$ tag to the C-terminus. The β1-LT subunit (Gene ID 25450) was synthesized and cloned into pEG BacMam by Bio Basic Inc. with residues K309-K414 replaced with a Gly-Thr linker, along with a thrombin site and a $His_8$ tag at the C-terminus. A bicistronic construct called $α1β1_{EM}$ containing α1-LT and β1-LT, in the context of the pEG BacMam vector, was used for large scale expression. The $γ2S_{EM}$ subunit construct has a three-residue (Gly-Arg-Ala) insertion between Asp374 and Cys375 along with a thrombin site, a $His_8$ tag and a 1D4 tag at the C-terminus. For electrophysiology experiments the FusionRed fluorophore was inserted in the γ2S subunit within the M3/M4 loop using an AscI restriction site; this construct is called γ2S-FR. A cartoon depiction of the constructs is provided in *Figure 1—figure supplement 1*.

## Preparation of an α subunit-specific Fab

The purified $α1β1_{EM}$ receptor complex in lauryl maltose neopentyl glycol was used for immunization. Initial screening returned a large number (>200) of IgG positive mAb candidates. ELISA screening was used to determine the highest binding antibodies against receptor in the native and denatured states at a 1:300 dilution. Those that selectively bound to the native state were chosen for further characterization. Subsequent ELISA binding results at a 1:3000 dilution further filtered the candidate pool to 40 supernatants which were examined by western dot blot. Subsequent FSEC analysis identified 8E3 as a α1 specific antibody (*Figure 1—figure supplement 2*). Isolated mAb at 0.5 mg/ml in 50 mM $NaPO_4$ pH 7 buffer containing 10 mM EDTA, 10 mM L-Cys, 1:100 papain (w/w) was digested for 2 hr at 37°C. The reaction was quenched by adding 30 mM Idoacetamide and placing on ice for 20 mins in the dark. After verification of the digestion by SDS page, the Fc portion was separated using Protein A resin and the flow through containing Fab was collected and concentrated for further use.

## Expression and purification

P1 virus for the $α1β1_{EM}$ and the $γ2S_{EM}$ was generated and amplified using standard procedures to obtain P2 virus. Viruses were stored at 4°C in the dark and supplemented with 1% FBS. Virus titer was determined using flow cytometry in a ViroCyt Virus Counter 2100 *Ferris et al., 2011Ferris et al., 2011Ferris et al., 2011Ferris et al., 2011*(*Ferris et al., 2011*). The tri-heteromer ($α1β1γ2S_{EM}$) was expressed in TSA201 suspension cells. Infection was performed at a cell density of $1.5-3 \times 10^6$ cells/mL in Gibco Freestyle 293 Expression medium supplemented with 1% FBS and placed in a humidity- and $CO_2$-controlled incubator. The total volume of virus added was less than 10% of the culture volume in all cases. Cells were infected with an MOI of 2 for both the viruses. After 24 hr of infection at 30°C, sodium butyrate and picrotoxin were added at final concentrations of 10 mM and 12.5 µM, respectively, and the flasks were shifted to 27°C for another 24 hr. All

procedures thereafter were done either on ice or in a cold room. Cells were harvested by centrifugation and the pellets were washed with TBS pH 8 containing 25 mM $MgCl_2$ and 5 mM $CaCl_2$. The pellets were re-suspended in 30 mL/L of culture volume in wash buffer with 1 mM PMSF, protease inhibitors (2 µg/ml leupeptin, 2.0 µM pepstatin and 0.8 µM aprotinin), 1.5 mM GABA, 20 µM ivermectin, and 25 µg/mL DNAse I. Cells were then sonicated for a 3:30 min cycle (15 s on/off) while stirring. Cell debris was removed by centrifugation at 7500 g for 20 mins, followed by centrifugation at 125,000 g for 1.5 hr, to pellet the cellular membranes. Membranes were re-suspended (~7 ml/lit culture) in 20 mM $NaPO_4$ pH 8, 200 mM NaCl, 1 mM $MgCl_2$, 1 mM $CaCl_2$, 5 µg/mL DNAse I, 0.3 mM PMSF, 1.5 mM GABA and 10 µM ivermectin and mechanically homogenized. Membranes were solubilized by adding 2% (w/v) C12M and 1 mM CHS for 1 hr at 4°C. Solubilized membranes were centrifuged for 50 min at 125,000 g. The supernatant was then mixed with 1D4 affinity resin equilibrated in 20 mM $NaPO_4$ pH 8, 200 mM NaCl, and 1 mM C12M for 3–4 hr at 4°C with gentle mixing. The resin was washed with 100 column volumes of the equilibration buffer; elution was achieved using a buffer supplemented with 0.2 mM 1D4 peptide. The eluted protein was concentrated by ultrafiltration, followed by the addition of 2.1 molar fold excess of 8E3 Fab. The concentrated sample was loaded onto a Superose 6 increase 10/300 GL column equilibrated with 20 mM HEPES pH 7.3, 200 mM NaCl, 1 mM C12M and 1.5 mM GABA. The flow rate was kept at 0.5 mL/min.

### Radio ligand binding assay

Binding assays were performed with the 10 nM $\alpha1\beta1\gamma2S_{EM}$ receptor-Fab complex using either 0.75–500 nM [³H]-muscimol or 1–1000 nM [³H]-flunitrazepam in a buffer containing 20 mM Tris pH 7.4, 150 mM NaCl, 1 mM C12M, 20 µg/ml BSA and 1 mg/ml YiSi Copper HIS TAG scintillation proximity assay beads (Perkin Elmer, MA). Flunitrazepam binding was measured in the presence of 1.5 mM GABA. Non-specific signal was determined in the presence of 1 mM GABA for the muscimol and 1 mM diazepam for flunitrazepam. Experiments were performed using triplicate measurements. Data was analyzed using Prism 7.02 software (GraphPad, CA) using a one site binding model.

### Electrophysiology

TSA-201 cells grown at 30°C in suspension were transfected with plasmid DNA encoding the bicistronic $\alpha1\beta1_{EM}$ construct and the monocistronic $\gamma2S$-FR construct using Lipofectamine 2000. Cells were plated on glass coverslips 2 hr prior to recording, and all recordings were conducted 18–36 hr after transfection.

Pipettes were pulled and polished to 2–4 M$\Omega$ resistance and filled with internal solution containing (in mM) 140 CsCl, 4 NaCl, 4 $MgCl_2$, 0.5 $CaCl_2$, 5 EGTA, 10 HEPES pH 7.4. Unless otherwise noted, external solutions contained (in mM) 140 NaCl, 5 KCl, 1 $MgCl_2$, 2 $CaCl_2$, 10 Glucose, 10 HEPES pH 7.4. For all electrophysiology experiments requiring Fab, 25 nM Fab was maintained in all bath and perfusion solutions. For potentiation experiments, currents were elicited via step from bath solution to solution supplemented with either 5 µM GABA or 5 µM GABA + 1 µM Diazepam. An unpaired t-test with Welch's correction was used to analyze changes in potentiation with or without Fab. External solution exchange was accomplished using the RSC-160 rapid solution changer (Bio-Logic). Membrane potential was clamped at −60 mV and the Axopatch 200B amplifier was used for data acquisition. All traces were recorded and analyzed using the pClamp 10 software suite.

### Cryo-EM data collection

Gold 200 mesh quantifoil 1.2/1.3 grids were covered with a fresh layer of 2 nm carbon using a Leica EM ACE600 coater. The grids were glow discharged at 15 mA for 30 s using Pelco easiGlow followed by the application of 4 µL 1 mg/mL PEI (MAX Linear Mw 40 k from Polysciences) dissolved in 25 mM HEPES pH 7.9. After 2 min, PEI was removed using filter paper immediately followed by two washes with water. The grids were dried at room temperature for 15 mins. Graphene oxide (Sigma) at 0.4 mg/mL was centrifuged at 1000 g for 1 min and applied to the grids for 2 mins. Excess graphene oxide was blotted away followed by two water washes. The grids were dried once again for 15 mins at room temperature before use. 2.5 µL of 0.15 mg/mL $\alpha1\beta1\gamma2S_{EM}$ receptor sample was applied to the grids with blot-force of 1 for 2 s using FEI Vitrobot in 100% humidity.

Grids were loaded into a Titan Krios microscope operated at 300 kV. Images were acquired on a Falcon three direct-detector using counting mode at a nominal magnification of 120,000,

corresponding to a pixel size of 0.649 Å, and at a defocus range between −1.2 to −2.5 μm. Each micrograph was recorded over 200 frames at a dose rate of ~0.6 e−/pixel/s and a total exposure time of 40 s, resulting in a total dose of ~37 e−/Å$^2$.

## Cryo-EM data analysis

A total of 1391 movie stacks were collected and correction for beam-induced motion was carried out using MotionCor2 (*Zheng et al., 2017*) (*Figure 1—figure supplement 4*). The dose weighted micrographs were used for determination of defocus values by Gctf (*Zhang, 2016*). Micrographs with large areas of ice contamination, multiple layers of graphene oxide, defocus values larger than −2.5 μm or defocus values smaller than −1.2 μm were deleted. Thus 1097 'good' micrographs were retained for the following image analysis. A total of 183,040 particles were automatically picked by DoG-picker (*Voss et al., 2009*), binned by 2x, and imported into cryosparc (*Punjani et al., 2017*) for 2D classification. In order to populate the particle stack with receptor-Fab complexes, only classes with clear features and clear background were chosen. A total of 34,911 particles were retained and used to generate an initial model, which was then subjected to homogenous refinement using cryosparc. The stack of 34,911 particles and the model were employed for further processing using Relion (*Scheres, 2012*).

We also carried out reference-based particle picking from the 1097 'good' micrographs with RELION, using seven of the 2D class averages from the cryoSPARC processing described above to derive the templates (*Figure 1—figure supplement 4*). This process yielded 216,543 particles which then were binned by 2x and subjected to one round of 2D classification to remove the 'junk' particles, yielding a total of 62,844 particles. Prior to further 3D refinement, the 'DoG-picker' selected particles (34911) were combined with the template-based particles (62,844) and an in-house script was used to remove the duplicates, this has been deposited at https://github.com/craigyk/emtools (*Yoshioka, 2018*; copy archived at https://github.com/elifesciences-publications/emtools). A total of 68,793 particles remained after removal of duplicates.

The combined, 2x binned particles were then used for 3D refinement in RELION by applying a mask focusing on the receptor and the variable domains of the Fabs. Subsequent 3D classification, without alignment, yielded 6 classes (*Figure 1—figure supplement 4*). The 49,417 particles belonging to class 2 were selected for a final refinement by RELION using the receptor-variable domain mask, yielding a map at ~3.8 Å resolution (*Figure 1—figure supplement 5*). We refer to this map as the 'whole map'. The 'whole map' was sharpened using Localscale (*Jakobi et al., 2017*). To further refine the ECD of the receptor, another mask focused on the ECD of the receptor and the variable domains of the Fabs was created, omitting the transmembrane domain and the micelle. Unbinned particles belonging to class 2 and class 3 were re-extracted and used in refinement, along with the ECD-variable domain mask, yielding a map at ~3.1 Å resolution (ECD map; *Figure 1—figure supplement 5*). The ECD map was sharpened using Phenix (*Terwilliger et al., 2018*). Local resolution was estimated using blocres as implemented in Bsoft (*Heymann and Belnap, 2007*).

Overall estimations of the resolutions of the reconstructions were carried out by Fourier Shell Correlation (FSC) = 0.143 criterion analysis (*Scheres and Chen, 2012*). The local resolution for the whole map varies about from 3.5 Å to 7 Å (*Figure 1—figure supplement 5*) where the low resolution areas are in the flexible or disordered regions of the TMD and constant domains of the Fabs. The local resolution for ECD map varies from about 3 to 4 Å.

## Model building

Homology models for the α1, β1 and γ2S subunit were generated using SWISS-MODEL (*Biasini et al., 2014*). The 'initial model' for the pentamer was generated via rigid body fitting of the subunit models to the density map using UCSF Chimera (*Pettersen et al., 2004*). The high quality of ECD map facilitated building of the ECD structure by way of iterative cycles of manual adjustment in Coot (*Emsley and Cowtan, 2004*) and refinement using phenix (*Afonine et al., 2018*). After phenix refinement, the map correlation coefficient (CC) between the map and the ECD model was 0.80, indicative of a good fit between the ECD model and the ECD map.

To build the 'whole model' using the whole map, the TMD derived from the 'initial model' was combined with the ECD model using Coot. Due to the lower resolution of the density in the TMD, it was only possible to fit the helices to their associated density and some of the residues with large

side chains. $\alpha-$Helical secondary structure restraints were placed on the TMD helices throughout refinement. Due to the weak density associated with the loop between helices M3-M4 (M3/M4 loop) for all of the subunits, and with the M4 helices for the two α subunits and the γ subunit, the M4 helices together with the M3/M4 loop were not included in the whole model. After phenix refinement, the CC between the whole map model and whole map was 0.74. The final model has good stereochemistry as evaluated by MolProbity (*Chen et al., 2010*) (*Supplementary file 1*).

## Acknowledgements

We thank David Weiss for the gift of GABA$_A$ receptor constructs, Dan Cawley for monoclonal antibody production, Cynthia Czajkowski and Dennis Dougherty for helpful comments, Avinash Patel and Eva Nogales for the graphene oxide protocol, and Heidi Owen for assistance with manuscript preparation. We also acknowledge the use of the Multiscale Microscopy Center and the Exacloud supercomputer cluster at OHSU. This research was supported by the NIH (EG R01 GM100400). EG is an Investigator with the Howard Hughes Medical Institute.

## Additional information

### Funding

| Funder | Grant reference number | Author |
|---|---|---|
| National Institute of General Medical Sciences | R01 GM100400 | Eric Gouaux |
| Howard Hughes Medical Institute | | Eric Gouaux |

The funders had no role in study design, data collection and interpretation, or the decision to submit the work for publication.

### Author contributions

Swastik Phulera, Conceptualization, Data curation, Formal analysis, Investigation, Methodology, Writing—original draft, Writing—review and editing; Hongtao Zhu, Conceptualization, Software, Validation, Methodology, Writing—original draft, Writing—review and editing; Jie Yu, Conceptualization, Writing—original draft, Writing—review and editing; Derek P Claxton, Conceptualization, Writing—review and editing; Nate Yoder, Formal analysis, Methodology, Writing—original draft; Craig Yoshioka, Validation, Visualization, Methodology, Writing—review and editing; Eric Gouaux, Conceptualization, Resources, Supervision, Funding acquisition, Validation, Investigation, Writing—original draft, Project administration, Writing—review and editing

### Author ORCIDs

Swastik Phulera (iD) http://orcid.org/0000-0002-1272-7712
Hongtao Zhu (iD) http://orcid.org/0000-0003-1522-0500
Derek P Claxton (iD) http://orcid.org/0000-0002-1374-7176
Nate Yoder (iD) http://orcid.org/0000-0002-9017-0673
Eric Gouaux (iD) http://orcid.org/0000-0002-8549-2360

### Decision letter and Author response

Decision letter https://doi.org/10.7554/eLife.39383.034
Author response https://doi.org/10.7554/eLife.39383.035

## Additional files

### Supplementary files

• Supplementary file 1. Statistics of data collection, 3D reconstruction and model.
DOI: https://doi.org/10.7554/eLife.39383.022

• Supplementary file 2. Statistics of solvent accessible surface area.
DOI: https://doi.org/10.7554/eLife.39383.023

• Transparent reporting form
DOI: https://doi.org/10.7554/eLife.39383.024

## Data availability

EM density maps have been deposited to EMDB, and the structure of the triheteromeric receptor has been deposited to the PDB.

The following datasets were generated:

| Author(s) | Year | Dataset title | Dataset URL | Database, license, and accessibility information |
|---|---|---|---|---|
| Phulera S, Zhu H, Yu J, Yoshioka C, Gouaux E | 2018 | Cryo-EM structure of the benzodiazepine-sensitive alpha1-beta1-gamma2S tri-heteromeric GABAA receptor in complex with GABA (Whole map) | https://www.rcsb.org/structure/6DW0 | Publicly available at the RCSB Protein Data Bank (accession no. 6DW0) |
| Phulera S, Zhu H, Yu J, Yoshioka C, Gouaux E | 2018 | Cryo-EM structure of the benzodiazepine-sensitive alpha1-beta1-gamma2S tri-heteromeric GABAA receptor in complex with GABA (ECD map) | https://www.rcsb.org/structure/6dW1 | Publicly available at the RCSB Protein Data Bank (accession no. 6DW1) |
| Phulera S, Zhu H, Yu J, Yoshioka C, Gouaux E | 2018 | Cryo-EM structure of the benzodiazepine-sensitive alpha1-beta1-gamma2S tri-heteromeric GABAA receptor in complex with GABA (Whole map) | http://www.ebi.ac.uk/pdbe/entry/emdb/EMD-8922 | Publicly available at the Electron Microscopy Data Bank (accession no: EMD-8922) |
| Phulera S, Zhu H, Yu J, Yoshioka C, Gouaux E | 2018 | Cryo-EM structure of the benzodiazepine-sensitive alpha1-beta1-gamma2S tri-heteromeric GABAA receptor in complex with GABA (ECD map) | http://www.ebi.ac.uk/pdbe/entry/emdb/EMD-8923 | Publicly available at the Electron Microscopy Data Bank (accession no: EMD-8923) |

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
