## [Decision Letter]

Thank you for submitting your article "Cryo-EM structure of the benzodiazepine-sensitive α1β1γ2S heterotrimeric GABA_A_ receptor in complex with GABA" for consideration by *eLife*. Your article has been reviewed by Richard Aldrich as the Senior Editor, Kenton Swartz as the Reviewing Editor and three reviewers. The following individual involved in review of your submission has agreed to reveal his identity: Mark L Mayer (Reviewer #1).

The reviewers have discussed the reviews with one another and the Reviewing Editor has drafted this decision to help you prepare a revised submission.

Summary:

This manuscript presents a new GABA-bound cryo-EM structure of the rat α1β1γ2S triheteropentameric GABA_A_ receptor (GABAR) resolved at 3.1 – 3.8 Å. The new structure captures a GABA-bound channel structure in the presence of a Fab fragments attached to the α subunits. The structure also reveals a novel N-linked glycosylation on the α subunit that surprisingly lies inside the ECD channel vestibule, which may partially block ion permeation and regulate receptor subunit assembly. The M3-M4 cytoplasmic regions were removed from the α and β subunits in the construct used for cryo-EM experiments and thus, no structural details about this important region were obtained. The manuscript is clear and well written. The approach is valid and the quality of data is good. The work is exciting and provides one of the first high-resolution structures of a heteropentameric GABAR. The structure adds significant new details about intersubunit interactions and provides a template for understanding GABA and allosteric drug modulator actions. It provides additional insight and complements the recent cryo-EM structure of the human GABAR solved in the presence of GABA and a BZD site antagonist (3.9 resolution) that was published in Nature (June 28) from the lab of Ryan Hibbs. Both structures reveal the N-linked glycosylation in the ECD channel vestibule, and somewhat poorer resolution of the TMD domain suggesting that in the heteropentameric protein this domain is dynamic. Specifics of some intersubunit residue interactions and potential numbers of GABA binding sites are different between the structures, which due to their resolution and different ligands present is not surprising. Overall, the paper is a significant contribution to the field and we have no major concerns. Each of the reviewers has identified issues that can readily be addressed by carefully revising the text. For clarity, we refer you to the individual reviews below. There are no requests for additional experiments.

Reviewer #1:

This paper reports the first structure of a tri-heteromeric α1β1γ2S GABA_A_ receptor solved by single particle cryo-EM that is an exciting advance appropriate for *eLife* without major revision.

An α subunit selective monoclonal antibody Fab fragment was used to break quasi 5-fold symmetry and identify subunit arrangement in 2D class averages. The structure was solved to 3.1 to 3.8 Å overall resolution, allowing atomic models to be built and refined using density maps for the extracellular domain, but with substantially lower resolution for the α-helical TM domains that was impacted by preferred orientation which the authors were unable to overcome after exploring multiple conditions for sample preparation. Despite this, the paper represents a tour de force study of challenging membrane protein biochemistry, coupled with state of the art cryo-EM data processing, and substantially increases our understanding of pentameric ion channels and in particular GABA_A_ receptor structural biology. With one exception, all of the features reported here are confirmed in two recent papers on closely related GABA_A_ receptor structures, leaving zero doubt that the work is sound.

Key findings from the study include the first clues and structural information on mechanisms controlling assembly of heteromeric GABA_A_ receptors; discovery of an unanticipated glycosylation site that partially occludes the outer vestibule of the ion channel, for which compelling structural data is provided, and that is unique to GABA_A_ receptor α subunits; and insights into ligand binding. Unexpectedly, the strongest ligand density is found at a non-canonical site, which the authors interpret with reasonable caution.

Although many details revealed by the structure were anticipated by extensive prior mutagenesis studies coupled with functional and biochemical analysis, the direct visualization of these mechanisms is a major advance, and refreshingly the authors do a reasonable job of discussing the structure in context of prior work, although this could be improved. Also, appropriately acknowledged is the fact that it is unclear whether the present agonist bound structure represents an open, desensitized, or preactivated state, along with possible issues that could impact this, but as discussed below this needs elaboration.

I have no major issues with the paper, but the following point could be better addressed during revision. In subsection “Conformation and asymmetry of the TMD”, the authors note that sample preparation conditions could potentially impact the observed structural asymmetry in subunit arrangement around the central pore that differs from the more symmetric arrangement in prior homomeric and di-heteromeric pentameric ligand gated ion channel structures. This is a frequent caveat for studies of detergent solubilized membrane proteins extracted from the lipid bilayer, and two recent papers on closely α1β2γ2 (Hibbs et al., 2018) and α1β3γ2 (Aricescu lab, https://www.biorxiv.org/ content/early/2018/06/05/338343) GABA_A_ receptors directly address this, revealing alternative, symmetric TMD conformations (Hibbs) and major changes in TMD structure depending on lipids added during purification (Aricescu). I believe that the authors should explicitly state that it is possible that the observed structural asymmetry in the TMD may represent a non-native conformation.

Reviewer #2:

The GABA_A_ receptors are major inhibitory neurotransmitter receptors in the central nervous system, responding to GABA binding with opening of a chloride permeable channel. Their dysfunction is associated with conditions including epilepsy, schizophrenia, and autism. The GABA_A_ receptors are members of the diverse pLGIC family which has been the subject of considerable structural work across prokaryotic and eukaryotic systems, which has defined core structural features. In the central nervous system, the most prevalent GABA_A_ assembly bears 2x α subunits, 2x β subunits, and 1x γ subunit, arranged as β-α-β-γ-α when viewed extracellularly. Despite the centrality of this receptor to human biology, basic questions including the basis of tri-heteromeric assembly and GABA binding have been incompletely explored.

To address this gap in fundamental understanding of GABA_A_ biology, Phulera and colleagues pursued the structure of the tri-heteromeric rat α1β1γ2 isoform, with M3/M4 loop deletions in α1 and β1 subunits (α1β1γ2S_EM_). They heroically expressed and purified this receptor and raised the 8E3 antibody specific to the α1 subunit. Before pursuing the structure of the receptor in complex with Fab 8E3, the authors established that Fab binding does not hinder receptor activity, while noting some reduction of diazepam potentiation. The GABA-bound structure was solved with the receptor solubilized in C12M-CHS, on grids coated with graphene oxide. This condition gave a preferred orientation but permitted resolution to a reported 3.8 Å for the full receptor, or 3.1 Å for the ECD with masking. They use these results for analysis of the subunit arrangement, subunit interfaces, the GABA binding site, glycosylation sites, and to conclude that the receptor is in an agonist/allosteric modulator bound, activated-like state.

The study involves impressive biochemical work to provide a look at the organization of a biologically and clinically significant receptor. The α1β1γ2S_EM_ construct was functionally validated, the cryo-EM work was well done, and the structural interpretations seem reasonable. The observed glycosylations and speculation about how they influence receptor assembly is particularly exciting and adds to an important foundation for future interrogation. Overall, the text and figures are clear and professional and I have no major concerns.

Reviewer #3:

This manuscript presents a new GABA-bound cryo-EM structure of the rat α1β2γ2 triheteropentameric GABA_A_ receptor (GABAR) resolved at 3.1 – 3.8 Å. The new structure captures a GABA-bound channel structure in the presence of a Fab fragments attached to the α subunits. The structure also reveals a novel N-linked glycosylation on the α subunit that surprisingly lies inside the ECD channel vestibule, which may partially block ion permeation and regulate receptor subunit assembly. The M3-M4 cytoplasmic regions were removed from the α and β subunits in the construct used for cryo-EM experiments and thus, no structural details about this important region were obtained. The manuscript is clear and well written. The approach is valid and the quality of data is good. The work is exciting and provides one of the first high-resolution structures of a heteropentameric GABAR. The structure adds significant new details about intersubunit interactions and provides a template for understanding GABA and allosteric drug modulator actions. It provides additional insight and complements the recent cryo-EM structure of the human GABAR solved in the presence of GABA and a BZD site antagonist (3.9 resolution) that was published in Nature (June 28) from the lab of Ryan Hibbs. Both structures reveal the N-linked glycosylation in the ECD channel vestibule, and somewhat poorer resolution of the TMD domain suggesting that in the heteropentameric protein this domain is dynamic. Specifics of some intersubunit residue interactions and potential numbers of GABA binding sites are different between the structures, which due to their resolution and different ligands present is not surprising. Overall, the paper is a significant contribution to the field and I have no major concerns.

---

## [Author Response]

Reviewer #1:[…]I have no major issues with the paper, but the following point could be better addressed during revision. In subsection “Conformation and asymmetry of the TMD” the authors note that sample preparation conditions could potentially impact the observed structural asymmetry in subunit arrangement around the central pore that differs from the more symmetric arrangement in prior homomeric and di-heteromeric pentameric ligand gated ion channel structures. This is a frequent caveat for studies of detergent solubilized membrane proteins extracted from the lipid bilayer, and two recent papers on closely α1β2γ2 (Hibbs et al., 2018) and α1β3γ2 (Aricescu lab, https://www.biorxiv.org/ content/early/2018/06/05/338343) GABA_A_ receptors directly address this, revealing alternative, symmetric TMD conformations (Hibbs) and major changes in TMD structure depending on lipids added during purification (Aricescu). I believe that the authors should explicitly state that it is possible that the observed structural asymmetry in the TMD may represent a non-native conformation.

We agree with the reviewer that the structural asymmetry observed could be a result of the detergent employed in the experiments and may represent a non-native conformation. We therefore modified the text in subsection “Conformation and asymmetry of the TMD**”** to reflect this caveat.